# Multifrequency Driven Nematics

**Noureddine Bennis \*, Jakub Herman**  **, Aleksandra Kalbarczyk, Przemysław Kula and Leszek R. Jaroszewicz**

Faculty of Advanced Technologies and Chemistry, Military University of Technology, 2 gen. S. Kaliskiego St., 00-908 Warsaw, Poland; jakub.herman@wat.edu.pl (J.H.); aleksandra.kalbarczyk@wat.edu.pl (A.K.); przemyslaw.kula@wat.edu.pl (P.K.); leszek.jaroszewicz@wat.edu.pl (L.R.J.)

\* Correspondence: noureddine.bennis@wat.edu.pl

**Abstract:** Liquid crystals act on the amplitude and the phase of a wave front under applied electric fields. Ordinary LCs are known as field induced birefringence, thus both phase and amplitude modulation strongly depend on the voltage controllable molecular tilt. In this work we present electrooptical properties of novel liquid crystal (LC) mixture with frequency tunable capabilities from 100Hz to 10 KHz at constant applied voltage. The frequency tunability of presented mixtures shown here came from composition of three different families of rodlike liquid crystals. Dielectric measurements are reported for the compounds constituting frequency-controlled birefringence liquid crystal. Characterization protocols allowing the optimum classification of different components of this mixture, paying attention to all relevant parameters such as anisotropic polarizability, dielectric anisotropy, and dipole moment are presented.

**Keywords:** nematic; dual frequency nematic; dielectric anisotropy; optical modulation

---

## 1. Introduction

Spatial light modulators (SLMs) based on liquid crystals (LCs) require very specific combinations of physical properties of liquid crystal mixtures to operate efficiently, requiring different combinations of components. Multicomponents mixtures of LCs find widespread use in photonic applications. The reason for this, is the wide temperature range necessary for applications [1]. The more important physical properties of LCs mixtures include electric permittivities, optical refractive indices, elastic constants, and viscosities. In order to understand the dielectric properties of LCs mixtures, some knowledge of the behavior of each component of the mixture is necessary. Owing to their electrical and optical properties, they are very sensitive to external electric and magnetic fields and they play an important role in many types of photonic devices enabling the development of multiphase spatial light modulators (SLMs) that can perform high-resolution, dynamic, optical beam positioning as well as temporal and spatial beam shaping. The optical system of such as spatial light modulators can perform with extremely high-resolution having a very small pixel size of about a few μm. This small resolution makes the latest generation of microdisplays based on LCs useful for many applications. However, we shall be concerned with negative physical effects related to smaller pixel sizes because of the inherent elasticity of the liquid crystal (LC) material, the surface anchoring of the alignment layer, and the fringing field of the discrete electrodes' voltage distribution [2]. The electrical field that exists between a pixel turned on and its neighbor at a different voltage causes part of the liquid crystal molecules to adopt a tilt angle opposite to the one in the main part of the pixel [3]. In order to avoid the fringing field effect originating from different values of voltage distribution, a digital driving frequency such as a pulse width modulation is needed. In this case, equal voltage with a predetermined frequency should be applied to adjacent pixel electrodes. As well as the amplitude of applied voltage

could alter the reorientation of LC molecules, also its frequency may alter their response. Thus, we need to understand the most important information about the molecular dynamics of LCs in the presence of the alternating electric field. When a highly anisotropic molecules of LCs are found in an electric field their electric polarization depends on their orientation with respect to it. The frequency dependence can be described from the physical origin of the induced polarization of the molecules. This induced polarization interacts with an external electric field and the torque tries to align the long molecular axis parallel or perpendicular to the external electric field depending of the sign of the dielectric anisotropy. If the applied field varies with time, then the frequency dependence of the permittivity is an additional property of the LCs material under study. Thus at much higher frequency of the applied electric field, the molecular dipoles do not rotate fast enough to contribute to the dielectric response, therefore the phase shift between the electric field E and electric polarization occurs. This delay yields to the dielectric relaxation of the LC molecules, causing the dielectric dispersion and losses [4]. For the uniaxial nematic LCs there are two independent components of the electric permittivity, parallel $\varepsilon_{\parallel}$ and perpendicular $\varepsilon_{\perp}$ to the axis of symmetry. Both principal components of the dielectric permittivity tensor show different frequency dependencies and temperature dependencies as well [5–7]. Perpendicular component $\varepsilon_{\perp}$ exhibits the high frequency process which can be treated as an ordinary Debye dispersion of fluid systems which usually occurs in the microwave region. However, $\varepsilon_{\parallel}$ may exhibit in some cases an additional dispersion at much lower frequencies in the radio-frequency region. This low-frequency dispersion of $\varepsilon_{\parallel}$ is caused by orientational relaxation of the permanent dipole moment of the parallel component to the long molecular axis. Maier and Meier are the first who formulated these ideas more qualitatively by applying Onsager's theory of static polarization to nematic LCs [8]. Measurements of the dielectric properties of LCs give relaxation frequencies, which reflect dynamic processes involving a change in electric polarization that results from the chemical structure of the mesogen forming the LC material. Many mixtures of LCs exhibit a single dielectric relaxation even if the components individually have relaxations in different frequency ranges. However, in some mixtures of mesogens having significantly different structures, for example mixtures of two-ring and three-ring mesogens, two separated relaxations are sometimes detected [9]. The possibility of designing mixtures with very low relaxation frequencies is useful for a dual frequency addressing displays [10]. Such mixture is usually formed by a combination of many components such as molecules having large transverse dipole moment with $\Delta\varepsilon < 0$ and molecules with a large longitudinal dipole moment with $\Delta\varepsilon > 0$ [11]. The availability of a dielectric anisotropy of either sign depending on the frequency of the applied field allows the possibility to improve the dynamic response of electro-optical devices. One of the main drawbacks of Dual frequency liquid crystal (DFLC) was the high crossover frequencies; problems arising when high frequencies are applied (> 50 kHz) are higher cost in the driving circuit, larger heat dissipation, and stronger dielectric heating [12]. For this reason, the research on materials with low crossover frequencies led to novel materials [13–15]. In this work, we report on LCs mixture design by mixing three different families of LC compounds. Two groups of the components are dielectrically positive nematics, having different generated dipole moment, and one group of compounds is dielectrically neutral, which has been chosen as viscosity adjusting component. The resulting mixture meets the requirements of the large positive dielectric anisotropy LCs to be continuously controlled by low frequencies of the electrical applied voltage. The dielectric anisotropy of the proposed mixture goes to zero instead of being negative at high frequencies of the electric field. The mesomorphic properties of the three components of the mixture under study and electro-optical features in new formulated mixture are reported.

## 2. Influence of NLC Mixtures on Dielectric Relaxation

Particular semiempirical models have been regularly used in the literature to allow the prediction of the electrooptical parameters such as the dielectric anisotropy $\Delta\varepsilon$ and the birefringence $\Delta n$ of dielectric materials. These two parameters can be derived from single molecule parameters, such as the dipole moment μ and the polarizability α, using the Maier–Meier theory [8]. According to this

theory, the supramolecular parameter $\Delta\varepsilon$ can be connected to the principal component of the molecular polarizability and the value and angular position of permanent electric dipole moment $\mu$.

Dielectric relaxation in LC material is influenced by the molecular structure used in the liquid crystal mixture and also by other factors such as intermolecular interactions and the local viscosity and temperature. This dependence gives raise to the possibility of designing liquid crystal mixtures with very low relaxation frequencies of electric permittivity parallel $\varepsilon_\parallel$, which is useful for many applications [16]. Apart from traditional SLM, the unique properties of this LC mixture can be used in all kind of optical phase modulators recently reported, e.g., adaptive lenses [17], beam steering [18], correction of aberrations [19], 3D vision applications [19–22], novel aberrations correctors for rectangular apertures [23], microaxicon arrays [24], multioptical elements [25], hole-patterned microlenses [26], multifocal microlenses [27], high fill-factor microlenses [28], frequency controlled [29] microlenses, optical vortices [30], lensacons, and logarithmic axicons [31]. The LC mixtures used in this work have been designed by mixing three different families of LC compounds to meet the requirements of the LC to be controlled by low frequencies of the electrical applied voltage. We have formulated the mixture using the set of compounds belonging to three different groups (see Table 1).

**Table 1.** General molecular structures of compounds used to form the investigated nematic mixture and their weight %.

| Components | Chemical Structure | Weight % |
|:---:|:---:|:---:|
| I |  | 6.5 |
| II |  | 63.5 |
| III |  | 30 |

The investigated material is a nematic composition of three different families of rod like LCs. The chemical formulas and the bond dipole moments of different groups forming the multifrequency driven nematic are shown in Figure 1. These structures belong to fluorine substituted 4-[(4-cyanophenoxy)carbonyl]phenyl 4-alkylbenzoates [32] (component I), fluorine substituted alkyl-alkyl phenyl-tolanes [32–34] (component II), and alkyl-alkyl bistolanes [35,36] (component III). Each component is characterized by their strong nematogenic character.

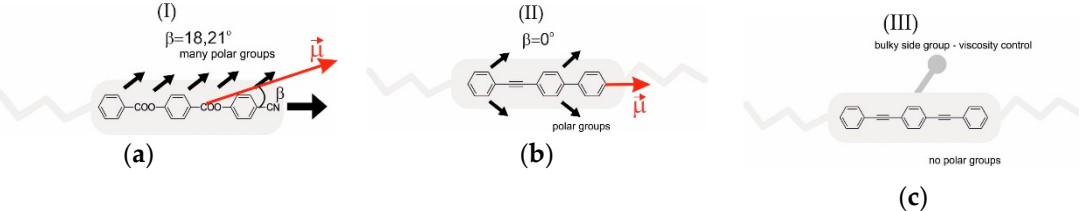

**Figure 1.** The structures of the three different groups forming the mixture: (**a**) component I, (**b**) component II and (**c**) component III. Red arrows indicate the position of the molecular dipole moments.

Two of the components are dielectrically positive nematics and one is dielectrically neutral. Components I and II have a large longitudinal dipole moment. Group I structures (Figure 1a) show a strong dipole moment that can be estimated as a vector sum of the single moments of all intramolecular chemical bonds. Their positions and orientations within the molecule for these dipole moments are

additives, having substantial influence on nematic ordering. The largest contribution to the dipole moment of component I comes from the cyano group. This in turn is coupled with two ester groups that additionally enhance dipole moment value. The longitudinal electric dipole moment of the component (I) is about 12,4 debye due to a triple −CN bond; the permanent dipole moment forms an angle with the molecular axis equal to 18.21°. This group of compounds is characterized by dielectric anisotropy $\Delta\varepsilon$ in the range from 60 to 80, which can be realized by the use of polar chemical groups, making this component a highly polar nematics liquid crystal. Additionally component I shows moderate molecular anisotropy and moderate electronic polarizability [36]. These highly polar LCs with high positive dielectric anisotropies tend to operate at lower voltages. Medium birefringent component II used in this mixture has a dipole moment of about 3 debye and $\Delta\varepsilon$ is in the range from +2 to +4 at 1 kHz. It has low melting enthalpies in order to enhance an excellent solubility with other two groups of components of the mixture. The group III is mainly used as diluters with which one may maintain the proper viscosity of LCs mixtures [32–34]. This component does not consist of polar substituents nor polar groups, therefore is a nonpolar highly birefringent liquid crystal. Bistolanes III show high anisotropy of molecular polarizability, as a result of a strongly conjugated system of benzene rings and carbon-carbon triple bonds [37]. In this component there is only an induced polarization that consists of two parts: the electronic polarization and the ionic polarization. Its dipole moment is μ=0. Thus, its $\Delta\varepsilon$ is expected to be small ($\Delta\varepsilon$ = +0,7 at 1 kHz) and is considered as medium to high birefringent with high nematogenity. This component is laterally substituted with short alkyl chains (methyl, ethyl, trifluoromethyl), shows strong nematic character, and therefore it minimizes the smectic formation in multicomponent systems under study. In contrast, in the LCs with polar molecules groups (I) and (II), there is, in addition to the total induced polarization, the orientation polarization due to the tendency of the permanent dipole moments to orient themselves parallel to the electric field.

## 3. Dielectric Measurements Results

In the context of the dielectric properties of the three components forming the multifrequency driven LCs mixture, their dielectric properties are of interest from a number of aspects. In this section we will focus on their dielectric relaxation spectra. the characterization involves the measure of the electric permittivity tensor over a range of frequencies from 100 Hz to 1 MHz. Dielectric investigation was carried out on the three components forming the mixture in planar aligned cells with thicknesses of 0.7 mm. The alignment was induced on the substrates by spin-coating polyimide SE-130 (Nissan Chemical Industries, Ltd., Tokyo, state, Japan), then baking at 180 °C for 1 h. The coated substrates were rubbed in the same direction and were assembled in antiparallel orientation. For this measurement, an external magnetic field was applied to the cell in order to orient the LCs molecules. In this case, two experimental geometries were applied; the parallel and perpendicular orientations of the director with respect to the electric field have been achieved by turning the electromagnet by 90° enabling the measurement of the $\varepsilon_{\parallel}$ and $\varepsilon_{\perp}$ which are parallel and perpendicular permittivity tensor components, respectively. The frequency dependences of the real (upper graphs) and imaginary (lower graphs) parts of the electric permittivity measured for the nematic phase of the three groups are shown in Figure 2.

The component (I) has relatively long molecules, therefore the resulting mesophases tend to be of high viscosity and are difficult to align; it has high melting and clearing temperatures and accordingly has very high viscosity which makes difficult the characterization of their dielectric properties. Therefore, the dielectric properties of this compound have to be measured in other neutral nematic matrices in order to extrapolate their resulting dielectric properties. In Figure 2a, only 10% molar concentration of component (I) has been used. The dielectric results of the pure material would have about 10 times higher $\Delta\varepsilon$, 10 times higher $\varepsilon_{\parallel}$, and more than 10 times lower relaxation frequency. For this component, the relaxation processes are detected at low frequencies and are more characteristic of collective modes normally associated with orientational relaxation of the permanent dipole moments. Components (II) and (III) have already much lower crystalline nematic transition temperatures, and therefore, they could be evaluated as pure systems. Those materials were

tested at 51 °C and 58 °C respectively. Under given temperatures, these components are far from nematic-isotropic transition, in consequence their properties within the nematic phase are flattened. The dielectric permittivity constants $\varepsilon_{\parallel}$ and $\varepsilon_{\perp}$ of the component III do not depend on frequency, therefore, dielectric losses are very small in the frequency range [100 Hz,15 KHz]. Figure 2 shows that the constituents of the formulated mixture have different dielectric responses to an external electric field of variable frequency. Note that the frequency dependence of the parallel component of electric permittivity is different for each group. We have formulated the mixture using the set of compounds belonging to three different families (See Table 2).

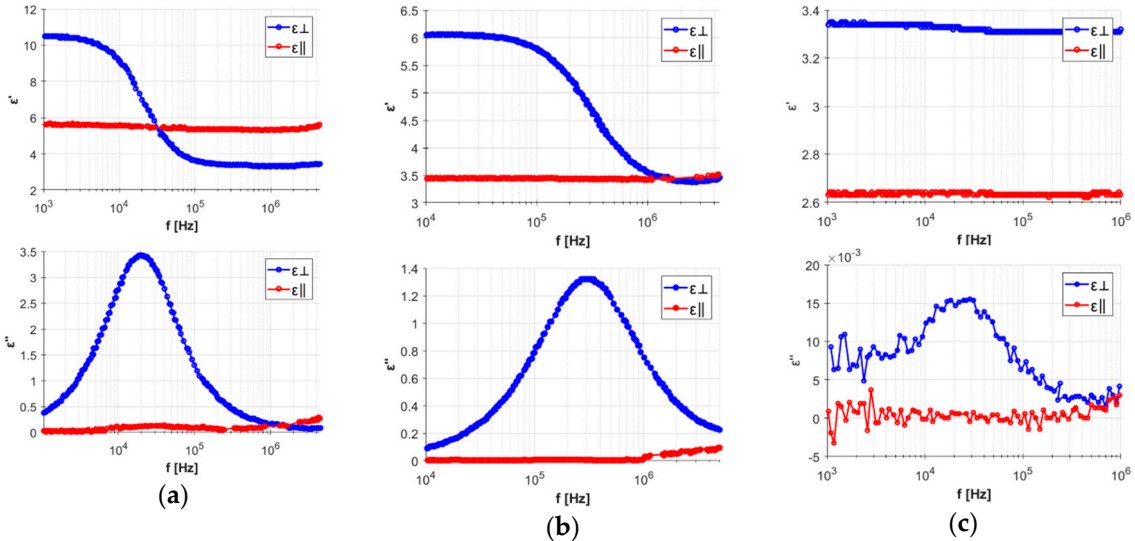

**Figure 2.** Frequency dependence of the real part (upper) and imaginary part (lower) of the complex permittivity of the formulated mixture in the measuring frequency range 100Hz to 1MHz for the three different components of the mixture: (**a**) component (I), (**b**) component (II) and (**c**) component (III).

**Table 2.** Mesomorphic properties of the three components of the mixture under study.

| Group | Polarity | Anisotropic Polarizability | Dipole Moment | Dielectric Anisotropy @ 1KHz | Viscosity |
|---|---|---|---|---|---|
| I | high | moderate | 12,4D | (60–80) | very high |
| II | moderate | high | 3D | (2–4) | low |
| III | very low | high | 0D | 0,7 | moderate |

Figure 3 shows the frequency dependence of the real and imaginary part of the complex permittivity of the formulated mixture in the measuring frequency range 100Hz to 1MHz. These results show that the mixing up of the three groups results in a positive nematic mixture with strong dependence on the frequency of the applied electric field. At a low frequency regime, the nematic mixture exhibits a large positive dielectric permittivity of about 9.5 and above 100Hz, $\varepsilon_{\parallel}$ starts to decrease linearly as frequency increases. However, $\varepsilon_{\perp}$ is kept constant at a value of 3.5. In the frequency range 100Hz to 1MHz, both $\varepsilon_{\parallel}$ and $\varepsilon_{\perp}$ stay at about 3,5. As a result, a continuously varying dielectric anisotropy, from 6 to 0 in the frequency range 100Hz-15KHz, is obtained. This mixture could be employed in optical devices which can readily be used as multiphase spatial light modulators controlled by a frequency.

The dielectric spectra showed in Figure 3a can be presented in the form of a Cole–Cole plot. This diagram represents a series of two interlinked semicircles whose radius of the semicircle are inversely proportional to the relaxation time, indicating two relaxation mechanisms separated in frequency. These results may then be explained as different dynamic modes associated with relaxation of different dipoles group, each relaxing at a different frequency. Assuming that, these contributions to the electric permittivity occur at sufficiently different frequencies, and therefore, they can be separated

in the dielectric spectrum. The dashed lines in Figure 3c mark two Debye-type processes fitted to the spectra. From those results, it seems that for our mixture two relaxation processes appear in the Cole–Cole plot corresponding to frequencies 3 KHz and 34 KHz, which show evidence of two or more separate relaxation processes.

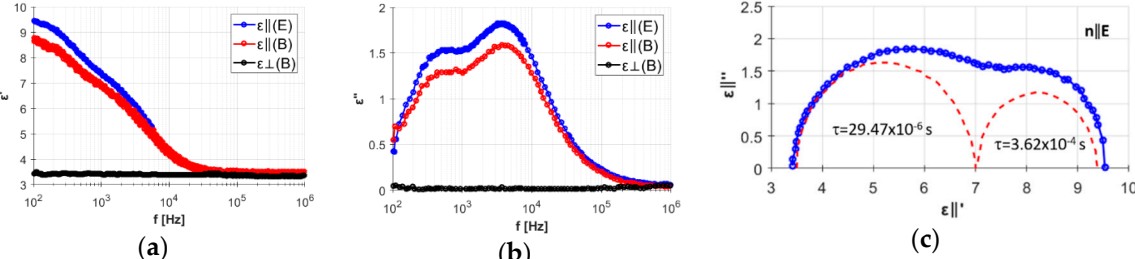

**Figure 3.** Frequency dependence of (**a**) real and (**b**) imaginary parts of the complex permittivity plotted in the measuring frequency range 100 Hz to 1MHz. (**c**) Cole–Cole plot showing relaxation process with two time constants.

Figure 4 shows the results of the optical retardance versus the alternative voltage waveform applied. For optical retardance measurements, we have used cells prepared with the same alignment method described above. However, the thickness of the assembled cells has been reduced 5 µm and next filled by the formulated mixture (Figure 4a) and by conventional liquid crystal 6CHBT (Figure 4b) when the voltage varies from 0 V to 15 V at a constant frequency. A maximal phase modulation depth obtained at $\lambda = 6323$ nm is $5\pi$ in case of 5005 and $2.7\,\pi$ in case of 6CHBT. Upon changing the frequency of the applied signal, a change in the retardance modulation is observed in the formulated mixture. This change is determined by the tunability of dielectric anisotropy by the frequency of the applied electric field. However, for LCs 6CHBT, since the relaxation frequency occurs in the 100 MHz region [7] no dependence of optical retardation on the frequency has been detected.

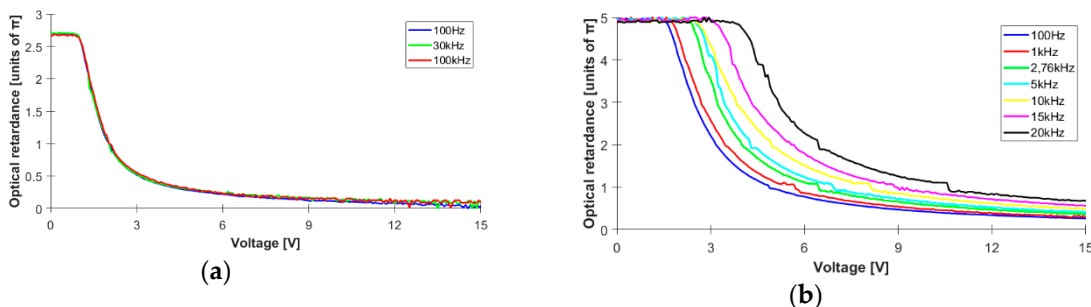

**Figure 4.** Voltage-dependent phase changes at different frequencies measured in 5 µm cell filled with: (**a**) formulated mixture, (**b**) 6CHBT liquid crystals.

These results show that as the frequency increases the electro-optical characteristics of the cell filled with multifrequency driven nematics shifts toward the higher operation voltage. The frequency dependence of the threshold voltage shown in Figure 4a reflects the frequency dependence of the dielectric anisotropy of the proposed LCs material. This frequency effect mainly originates from frequency dependence of dielectric permittivity $\Delta\varepsilon$ since all other parameters are independent of frequency.

## 4. Conclusions

Dielectric and electrooptical measurements were carried out on the novel liquid crystal mixture with frequency tunable capabilities from 100 Hz to 10 KHz. The dielectric anisotropy of the proposed mixture goes to zero at higher frequencies instead of being negative, which is typical for dual frequency nematics (DFN). Characterization protocols allowing the optimum classification of different components

of this mixture, paying attention to all relevant parameters such as anisotropic polarizability and dielectric anisotropy have been presented. The dielectric permittivity of the mixtures based on the three components depends upon the polarity of each particular group and their concentrations in the mixture. These studies can aid the development of new liquid crystals materials with carefully tailored dielectric properties.

**Author Contributions:** Methodology, writing and editing, N.B.; optical retardance measurement, A.K.; synthesis of the liquid crystalline single compounds and formulation of the mixture, J.H.; discussion of the results, P.K.; and supervision of the work, L.R.J.

**Funding:** Ministry of National Defense of Poland under grant number (GBMON/13-995/2018/WAT), Military University of Technology grant no 23-895.

**Acknowledgments:** The authors would like to thank Stanisław Urban for the dielectric measurements and his helpful discussions.

**Conflicts of Interest:** The authors declare no conflict of interest.

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
