# Peer review of "Multifrequency Driven Nematics"

_crystals, doi:10.3390/cryst9050275_

Round 1

Reviewer 1 Report

The paper contains very interesting results and it is well written, worthwhile to be published.

Some minor revisions are required.

Figure qualities need to be improved.

Please indicate what a b c are in figure captions of Fig. 2.

Author Response

We thanks the reviewer for his thorough revision of our manuscript. Please, find below our answer to his comments and a description of the changes made in the manuscript accordingly. We believe that, following reviewer suggestions,  has improved our manuscript

Reviewer comment: Figure qualities need to be improved

Answer: The figure has been improved

Reviewer comment: Please indicate what a b c are in figure captions of Fig. 2.

Answer: We have indicated in the caption of Figure 2, the description of (a), (b) and (c).

Reviewer 2 Report

The comments are attached in the pdf file.

Author Response

We thanks the reviewer for his thorough revision of our manuscript. Please, find below our answer to his comments and a description of the changes made in the manuscript accordingly. We believe that, following reviewer suggestions,  has improved our manuscript

Reviewer comment: This paper describes a method of making a new type of LC material. Related to the change in delta epsilon with frequency, the effect of each of the three materials is independently described. Overall the work is novel, timely and of broad interest. For these reasons I am supportive of publication. However, I think the manuscript would be significantly improved if the following points could be addressed.

1. Apart from SLM, this novel LC mixture could be useful in several applications. I recommend to introduce or conclude the manuscript with some phase only applications, recently published, in which this new material could be useful.

Answer 1: We would like to thanks the reviewer for his recommendations. To highlight some phase only applications in which our proposed mixture could be employed we have this paragraph that contains references recently published work, in which this new mixture could be useful.

Apart from traditional SLM, the unique properties of this LC mixture can be used in all kind of optical phase modulators recently reported, e.g adaptive lenses [17], beam steering [18], correction of aberrations [19], 3D vision applications [1922], novel aberrations correctors for rectangular apertures [23], micro-axicon arrays [24], multi-optical elements [25], hole-patterned [26], multi-focal [27], high fill-factor [28] and frequency controlled [29] microlenses, optical vortices [30], lensacons, logarithmic axicons [31].

2. It could be help for the reader to remark differences and advantages over conventional dual frequency LCs.

Answer 2: As commented in the manuscript, the ordinary parameter (εo) has a Debye type dispersion typical of fluids. This one usually occurs at microwave frequencies (>109Hz). The extraordinary parameter (εe) usually shows a lower frequency dispersion, in the radio frequency range. The transition of this frequency usually is very abrupt for frequencies around 100kHz. In dual frequency LC, this transition is in a more workable range than typical nematic LC, around 50 kHz. The applications of this material are based on use the two different dielectric anisotropies, positive (e.g 10 kHz) and negatives (e.g 60 kHz), the transition is usually very abrupt.

As it is commented in [13], the main problem of DFLC is the high crossover frequencies, a technical challenge for the application of DFLC when working at high frequencies (> 50 kHz) is that a too high operating frequency leads to (1) higher cost in driving circuit design; (2) larger heat dissipation; and (3) stronger dielectric heating.

Therefore developing low crossover frequency DFLCs is a very urgent and critical task for its application. A DFLC mixture usually consists of two types of compounds: (1) positive compounds whose dielectric anisotropy (∆ε) is positive at low frequencies but decreases as the driving frequency increases because of dielectric relaxation; and (2) negative compounds whose ∆ε is always negative and stays almost constant when the driving frequency is below the MHz range.

In the referred work they have developed low frequency crossover DFLC. The crossover frequency is more than 10 KHz and the whole transition occurs in one decade or less, the transitions are very abrupt.

In the case of the presented LC, all components of this mixture are positive or dielectrically neutral, in this case there is only positive dielectric anisotropy. The novelty of this LC is a smooth and continuous dielectric anisotropy from a determined value to zero in more than two decades (see Figure 15b, 100 Hz to 15 kHz). Moreover, the positive dielectric anisotropy in the whole range could be useful for some photonic applications.

In order to highlight this point, the following text has been included in the introduction section:

One of the main drawbacks of DFLC was the high crossover frequencies, problems arising when high frequencies are applied (> 50 kHz) are higher cost in the driving circuit; larger heat dissipation; and stronger dielectric heating [13]. For this reason, the research on materials with low crossover frequencies led to novel materials [14] [15].

3. There are some minor grammatical errors:

Answer3: we appreciate the grammatical corrections found by review: corrections done.

Reviewer 3 Report

The work presents a novelty mixture of 3 different LC molecules in order to incorporate a frequency tunable characteristics to the LC devices.

First of all, figures axes and labels are too small and it is not easy to read, please consider to make them more legible.

In "Dielectric measurements results" I would like to know more about how the samples have beenn aligned, please consider to include more details about the alignment method.

In Fig. 4 when the mixture is compared with the "conventional" 6CHBT it is not clear if the authors are using the same planar aligned cells of 0.7mm or are using a comercial cell. It is important to compare in the same conditions.

Author Response

We thanks the reviewer for his thorough revision of our manuscript. Please, find below our answer to his comments and a description of the changes made in the manuscript accordingly. We believe that, following reviewer suggestions,  has improved our manuscript

Reviewer comment: The work presents a novelty mixture of 3 different LC molecules in order to incorporate a frequency tunable characteristics to the LC devices.

First of all, figures axes and labels are too small and it is not easy to read, please consider to make them more legible.

Answer: We have improved the figures

Reviewer comment: In "Dielectric measurements results" I would like to know more about how the samples have been aligned, please consider to include more details about the alignment method.

Answer: We have added in the text this paragraph (see line 150,151 and 152)

 “The alignment was induced on the substrates by spin-coating polyimide SE-130 (Nissan Chemical Industries, Ltd), then baking at 180°C for 1 h. The coated substrates were rubbed in the same direction and were assembled in antiparallel orientation.

Reviewer comment: In Fig. 4 when the mixture is compared with the "conventional" 6CHBT it is not clear if the authors are using the same planar aligned cells of 0.7mm or are using a commercial cell. It is important to compare in the same conditions.

Answer: For optical retardance measurements we have used cells prepared with the same alignment method described above. However the thickness of the assembled cells has been reduced to 5 mm and next filled by the formulated mixture (Figure 4(a)) and by conventional liquid crystal 6CHBT Figure 4(b), when the voltage vary from 0V to 15 V at constant  frequency

Reviewer 4 Report

I this experimental work the authors deal with preparation and characterization of nematic liquid crystal (LC) mixtures with frequency dependent dielectric anisotropy. They focus on three compounds with different dispersions of the dielectric response, determine the appropriate mixing ratio, and study the dielectric and optical properties of the mixture.

As the most modern LC displays rely on electro-optic switching of nematic LCs, specific compounds having frequency-dependent dielectric anisotropy are actively studied all over the world. The main advantage of such so-called dual-frequency nematics is that they can be realigned either perpendicularly or parallel to the electric field by applying alternating voltage of different frequency to the same systems of electrodes.  Throughout decades, various corresponding LC mixtures have been proposed and characterized (see e.g. Refs. 10, 15, 18), which typically exhibit positive anisotropy at lower frequencies of the driving electric field and negative anisotropy at higher frequencies.

In this context, the authors here formulate their main achievement as the fact that the dielectric anisotropy of their mixture tends strictly to zero at higher frequencies. Despite the apparent formal novelty, the practical importance of this fact, however, remains uncertain. Although they discuss in the Introduction some practical problems arising in microdiplay design, I could not trace the connection with the high-frequency limit of the anisotropy.

There are also certain technical issues related to the figures. First, the marginal graphic quality and the tiny fonts make Figures 2-4 barely comprehensible. Second, the captions are not informative and, as in Fig.2, even erroneous.

Author Response

We thanks the reviewer for his thorough revision of our manuscript. Please, find below our answer to his comments and a description of the changes made in the manuscript accordingly. We believe that, following reviewer suggestions,  has improved our manuscript

Reviewer comment: In this context, the authors here formulate their main achievement as the fact that the dielectric anisotropy of their mixture tends strictly to zero at higher frequencies. Despite the apparent formal novelty, the practical importance of this fact, however, remains uncertain.

Answer: A DFLC mixture usually consists of two types of compounds: (1) positive compounds whose dielectric anisotropy (∆ε) is positive at low frequencies but decreases as the driving frequency increases because of dielectric relaxation; and (2) negative compounds whose ∆ε is always negative and stays almost constant when the driving frequency is below the MHz range. As it is commented in [13], the main problem of DFLC is the high crossover frequencies, a technical challenge for the application of DFLC when working at high frequencies (> 50 kHz) is that a too high operating frequency leads to (1) higher cost in driving circuit design; (2) larger heat dissipation; and (3) stronger dielectric heating. In the case of the mixture presented in this work, all components of this mixture are positive or dielectrically neutral, in this case there is only positive dielectric anisotropy. The novelty of this LC is a smooth and continuous dielectric anisotropy from a determined value to zero in more than two decades (see Figure 15b, 100 Hz to 15 kHz). Moreover, the positive dielectric anisotropy in the whole range could be useful for some photonic applications. In order to highlight this point, the following text has been included in the introduction section:

One of the main drawbacks of DFLC was the high crossover frequencies, problems arising when high frequencies are applied (> 50 kHz) are higher cost in the driving circuit; larger heat dissipation; and stronger dielectric heating [13]. For this reason, the research on materials with low crossover frequencies led to novel materials [14] [15].

The references [13] [14] and [15] are added to the manuscript

Reviewer comment: Although they discuss in the Introduction some practical problems arising in microdiplay design, I could not trace the connection with the high-frequency limit of the anisotropy. 

Using a LC material that can be controlled by  frequency, will help to solve some arising problems from microdisplay based on LC. To clarify this point we added in the introduction this sentence” In order to avoid the fringing field effect originating from different value of voltage distribution, digital driving driving frequency such as a pulse width modulation is needed. In this case, equal voltage with a predetermined frequency should be applied to adjacent pixel electrode.”

Reviewer comment: There are also certain technical issues related to the figures. First, the marginal graphic quality and the tiny fonts make Figures 2-4 barely comprehensible. Second, the captions are not informative and, as in Fig.2, even erroneous.

Answer: The figure quality has been improved and We have indicated in the caption of Figure 2, the description of (a), (b) and (c).
